# Impact of MYC on Anti-Tumor Immune Responses in Aggressive B Cell Non-Hodgkin Lymphomas: Consequences for Cancer Immunotherapy

**DOI:** 10.3390/cancers12103052

**Published:** 2020-10-20

**Authors:** A. Vera de Jonge, Tuna Mutis, Margaretha G. M. Roemer, Blanca Scheijen, Martine E. D. Chamuleau

**Affiliations:** 1Department of Hematology, Amsterdam UMC, VU University Medical Center, Cancer Center Amsterdam, 1081HV Amsterdam, The Netherlands; t.mutis@amsterdamumc.nl (T.M.); m.chamuleau@amsterdamumc.nl (M.E.D.C.); 2Department of Pathology, Amsterdam UMC, VU University Medical Center, Cancer Center Amsterdam, 1081HV Amsterdam, The Netherlands; m.roemer@amsterdamumc.nl; 3Department of Pathology, Radboud UMC, Radboud Institute for Molecular Life Sciences, 6525GA Nijmegen, The Netherlands; blanca.scheijen@radboudumc.nl

**Keywords:** MYC, diffuse large B cell lymphoma, high grade B cell lymphoma, tumor immune evasion, MYC inhibition, immunotherapy, T cell therapy

## Abstract

**Simple Summary:**

The human immune system has several mechanisms to attack and eliminate lymphomas. However, the MYC oncogene is thought to facilitate escape from this anti-tumor immune response. Since patients with MYC overexpressing lymphomas face a significant dismal prognosis after treatment with standard immunochemotherapy, understanding the role of MYC in regulating the anti-tumor immune response is highly relevant. In this review, we describe the mechanisms by which MYC attenuates the anti-tumor immune responses in B cell non-Hodgkin lymphomas. We aim to implement this knowledge in the deployment of novel immunotherapeutic approaches. Therefore, we also provide a comprehensive overview of current immunotherapeutic options and we discuss potential future treatment strategies for MYC overexpressing lymphomas.

**Abstract:**

Patients with MYC overexpressing high grade B cell lymphoma (HGBL) face significant dismal prognosis after treatment with standard immunochemotherapy regimens. Recent preclinical studies indicate that MYC not only contributes to tumorigenesis by its effects on cell proliferation and differentiation, but also plays an important role in promoting escape from anti-tumor immune responses. This is of specific interest, since reversing tumor immune inhibition with immunotherapy has shown promising results in the treatment of both solid tumors and hematological malignancies. In this review, we outline the current understanding of impaired immune responses in B cell lymphoid malignancies with MYC overexpression, with a particular emphasis on diffuse large B cell lymphoma. We also discuss clinical consequences of MYC overexpression in the treatment of HGBL with novel immunotherapeutic agents and potential future treatment strategies.

## 1. Introduction

Non-Hodgkin lymphomas (NHL) are the most common hematologic malignancies and can be divided into indolent and aggressive subtypes [1]. Diffuse large B cell lymphoma (DLBCL) is the most common aggressive NHL subtype with an incidence of 695 per 100,000 [2]. The outcome of patients with DLBCL is heterogeneous and dependent on clinical and biological variables. Currently, the only prognostic marker that has direct therapeutic implications for first-line therapy is overexpression of the *c-MYC* oncogene (hereafter *MYC*) [3]. MYC overexpression is either due to chromosomal translocation, gene amplification or somatic (hyper)mutations [4]. About 5–15% of all DLBCLs harbor translocations that affect the *MYC* gene located on chromosome 8q24.21 as established by fluorescence in situ hybridization (FISH) [5]. Translocation partners involve the enhancer of the immunoglobulin (Ig) heavy chain [t(8;14)], Ig lambda light chain [t(8;22)], and Ig kappa light chain genes [t(2;8)] or non-Ig gene regulatory elements [6]. In about 30% of the *MYC* translocated DLBCL patients, this is the only translocation (single hit (SH) DLBCL), while in the majority *MYC* translocations are accompanied by a translocation affecting either the *BCL2* or *BCL6* gene, referred to as double-hit (DH) high grade B cell lymphoma (HGBL), or both *BCL2* and *BCL6* genes, referred to as triple hit (TH) HGBL [1]. Concurrent overexpression of the MYC and BCL2 protein without underlying evidence for gene translocations is known as a “double-expressor” (DE) lymphoma [7]. Recent studies showed that HGBL with specific gene expression signatures (double hit signature (DHITsig) or molecular high-grade (MHG)), were enriched for, but did not exclusively contain, SH, DH or TH HGBLs [8,9]. In this review, we refer to both SH, DH or TH HGBL and DE lymphomas with “MYC overexpression”, since this eventually all results in high MYC protein expression.

Over the past decades, the clinical outcome of B cell NHL patients significantly improved with the introduction of immunotherapy by targeting cell surface molecules, such as CD20, with monoclonal antibodies [10]. However, progression free survival and overall survival are poor in patients with *MYC* translocations after treatment with standard immunochemotherapy for DLBCL (rituximab, cyclophosphamide, doxorubicin, vincristine and prednisone (R-CHOP)) [11,12,13,14,15,16]. Therefore, patients with DH and TH HGBL are often treated with dose-intensification regimens, such as dose-adjusted etoposide, prednisone, vincristine, cyclophosphamide, doxorubicin and rituximab (DA-EPOCH-R) [17]. Patients with SH and DE lymphomas have a prognosis in between DLBCL patients without MYC overexpression and patients with DH or TH HGBL [16,18]. Treatment strategies are usually not adapted for SH and DE lymphoma patients.

In recent years, numerous novel immunotherapeutic strategies have been tested in patients with B cell NHL. This includes immune checkpoint inhibitors, bispecific antibodies and CAR-T cell therapies [19]. To deploy these novel immunotherapeutic strategies in MYC overexpressing lymphoid malignancies, it will be important to understand the effects of MYC overexpression on anti-tumor immune responses. In this review, we highlight current understanding of impaired immune responses in MYC overexpressing lymphoid malignancies with particular emphasis on DLBCL. Preclinical data are illustrated by Burkitt lymphoma (BL; a rare subtype of NHL with a specific morphology and characterized by *MYC* translocation in 95–99% of the cases) models [20]. Furthermore, we provide a comprehensive overview of advanced developments in immunotherapeutic strategies for MYC overexpressing lymphoid malignancies.

## 2. The Role of MYC in Normal B Cell Development

MYC is a basic-helix-loop-helix leucine-zipper (bHLH-LZip) nuclear protein that forms a heterodimer with MYC associated factor X (MAX). By binding to a specific DNA sequence, the CACGTG E-box [21], the MYC/MAX heterodimer regulates transcription of 10–15% genes, that are involved in essential biological processes, such as cell growth, proliferation, differentiation, metabolism, stemness, apoptosis and protein translation [22,23,24,25]. As such, MYC regulates the development and maturation of lymphocytes [24,26,27,28,29].

Normal B cells develop from a hematopoietic stem cell via lymphoid progenitor cells into an early pro-B cell, pro-B cell, pre-B cell and, finally, an immature B cell. In the transition from the pro-B cell to pre-B cell stage, the pre-B cell receptor (BCR) is expressed, a process that is associated with MYC upregulation [30,31,32]. MYC promotes B cell proliferation and differentiation by activating several B cell determining genes, such as *Cd19* [33,34,35,36,37]. Immature B cells expressing a complete, functional, non-autoreactive BCR downregulate MYC expression, enter the peripheral circulation and then accumulate in lymph nodes where further maturation takes place under differential MYC expression [38,39]. MYC is re-expressed after (antigen) activation of the BCR, promoting proliferation and inhibiting differentiation of mature B cells [39,40]. After positive selection, MYC expression is downregulated, allowing B cells to differentiate into memory B cells or plasmablasts and to exit the germinal center [39,41]. In plasmablasts, MYC expression is suppressed, allowing terminal B cell differentiation into plasma cells [42].

## 3. The Role of MYC Overexpression on the Immune System in Lymphoid Malignancies

### 3.1. Impact of MYC Overexpression for Adaptive Immunity

Tumors possess the ability to evade immune detection and cytotoxic T cell responses by downregulating antigen presenting and costimulatory molecules and by inducing T cell tolerance. Research over the past decades has clearly demonstrated the important impact of MYC overexpression on the antigen-specific cross-talk between tumor cells and T cells. The negative influence of MYC on the adaptive immune system occurs at several levels as outlined below. A schematic summary of the effects of MYC overexpression on adaptive and innate anti-tumor immune responses is shown in Figure 1, and a detailed description follows in the text below.

#### 3.1.1. The Effect of MYC Overexpression on Antigen Presentation

Evasion of immune surveillance in lymphomas overexpressing MYC was first described in 1985, when Epstein-Barr virus (EBV) positive B cell NHL cell lines harboring an *MYC* translocation were found insensitive to cytotoxic T cell responses, irrespective of antigen expression [43]. Subsequent studies revealed that MYC overexpressing B cell NHL cell lines downregulates expression of HLA class I molecules, leading to impaired T cell recognition [44,45]. This phenomenon has also been widely observed in solid tumors [46,47,48,49,50].

The first clear evidence that *MYC* also negatively influences HLA class II restricted T cells was provided in 2015 [51]. Inducible overexpression of MYC in a preclinical BL model diminished peptide presentation via HLA class II through decreased expression of class II editor HLA-DM. Conversely, inhibition of MYC elevated HLA-DM and partially restored antigen presentation to CD4+ T cells [51]. These findings were recently confirmed in large gene expression profiling studies of HGBL [8,9].

#### 3.1.2. Consequences of MYC Overexpression on the Expression of Adhesion and Costimulatory Molecules 

Intracellular adhesion molecules (ICAMs) and vascular cell adhesion molecule-1 (VCAM-1) are highly important for T cell recruitment and activation. ICAM-1 promotes T cell activation and migration upon binding with its receptor lymphocyte function-associated antigen-1 (LFA-1) [52]. In *MYC*-transfected B-LCL, *MYC* downregulates surface expression of adhesion molecules LFA-1 [53], CD54 (ICAM-1) and CD58 (LFA-3) [54]. A series of BL biopsies also had a uniformly low LFA-1 expression pattern [53]. NF-κB target genes, including VCAM-1, are downregulated by MYC overexpression in a B-LCL cell line [55]. MYC overexpression is also inversely correlated with tumor necrosis factor (TNF) super family members lymphotoxin-α(TNFSF1), lymphotoxin-β and TNF-α, that play a role in T cell recognition, cell-to-cell communication and adhesion in B cell lymphoma cell lines [56].

The regulation of immune responses is a well-balanced counter play between co-stimulatory and co-inhibitory signals. Costimulatory molecules provide important secondary signals upon ligation to augment T cell responses. One of the earliest reports on the influence of *MYC* on the expression levels of T cell costimulatory molecules was provided by conditional cell lines, where switching MYC off resulted in CD40 downregulation [57]. CD40, a TNF receptor on B cells, is a costimulatory molecule that binds to CD40L (CD154) on T cells derived from the TNF superfamily [58]. Furthermore, *MYC* repressed CD80 in both an EBV transformed cell line and a transgenic B cell lymphoma mouse model [55,57]. The immunoglobulin superfamily member CD80 binds to co-costimulatory molecules from the CD28/B7 family on T cells [59]. However, in a cohort of 211 de novo DLBCL patients MYC protein overexpression did not correlate with CD40 gene expression, but an association was shown with decreased levels of the co-stimulatory immune checkpoint *OX40* (*TNFRSF4*) gene, decreased T cell receptor signaling molecules and regulatory T cell gene *FOXP3* [60].

#### 3.1.3. The Role of MYC Overexpression on PD-L1-Mediated T Cell Tolerance 

Programmed death-ligand 1 (PD-L1), also known as *B7-H1* or *CD274*, is a member of the B7 receptor family and the primary ligand for PD-1, one of the most important co-inhibitory molecules on T cells to regulate excessive T cell responses [61]. In DLBCL, high PD-L1 expression in tumor cells correlates with the non-germinal center B cell (non-GCB) subtype [62,63,64,65,66,67,68], inferior treatment outcome [63,66] and decreased progression free survival [66] and overall survival [62,63,65,68].

In various solid tumors, MYC overexpression is correlated with PD-L1 surface expression [69,70,71,72,73,74] and PD-L1 mRNA expression [69,71,72,73]. However, the exact relationship between MYC and regulation of PD-L1 expression in lymphoid malignancies is remains unclear. For instance, one study showed a positive correlation between MYC protein and PD-L1 mRNA and protein levels in a cohort of 108 de novo DLBCL patients [67]. In agreement with this, a recent murine study demonstrated that *Myc* could directly bind to the promotor of *Pd-l1* and that switching off *Myc* expression downregulated *Pd-l1* [75]. This was also found in human DLBCL cell lines treated with MYC inhibitor 10058-F4 or siRNA against *MYC* [67].

However, not all studies could confirm a direct relation between MYC and PD-L1. Namely, while a murine study observed PD-L1 downregulation after treatment of *Myc* endogenous mice with bromodomain inhibitor JQ1, no PD-L1 downregulation was observed after *Myc* shRNA knockdown [76]. This suggested that decreased PD-L1 expression levels were merely due to other effects of JQ1, rather than Myc downregulation [76]. Another study showed that MYC inactivation could increase PD-L1 mRNA levels [77] and it was proposed that not MYC, but STAT1, directly binds to the regulatory region of *PD-L1*. Since MYC and STAT1 can regulate their mutual expressions [78,79], it seemed possible that MYC inactivation increased STAT1, leading to increased PD-L1 expression [77]. Furthermore, two other cohorts of de novo and relapsed DLBCL patients reported a negative correlation between MYC and PD-L1 [63,64]. Finally, some other studies were not able to detect any correlation between MYC and PD-L1 at all in either DLBCL cell lines [80], nor in DLBCL patients [60,66,68,81]. Notably, EBV infection causes B cells to escape from immune surveillance [82,83] partly by PD-L1 overexpression [84,85], adding further complexity to the relationship between MYC and PD-L1 in EBV-associated DLBCL and BL.

To conclude, these studies employing different technical approaches, showed conflicting outcomes on the exact correlation between MYC and PD-L1 and demonstrate a complex interplay of different factors regulating PD-L1 expression in lymphoid malignancies.

### 3.2. Impact of MYC Overexpression for Innate Immunity

According to the “missing-self” hypothesis, NK cells target infectious and malignant cells mainly upon downregulation of HLA class I on the target cell [86]. As described in the previous section, MYC downregulates HLA class I expression [8,43,44,45], which would be beneficial for NK cell activity. However, it was recently suggested that downregulation of HLA class I in vivo resulted in NK cell tolerance and that the missing-self hypothesis may not apply to all cells downregulating MHC [87]. Two studies further investigated consequences of MYC overexpression on NK cell-mediated cytotoxicity. Investigating the systemic effects of *Myc* in murine T cell lymphomas a recent study showed that *Myc* expression was associated with lower NKp46 + NK cells in circulation, lower NK cell maturation and more disease progression than control mice [88]. In contrast, another study reported that MYC mRNA levels in B cell lymphomas and solid tumors are positively correlated with B7-H6 expression on the tumor cell, a ligand for the NK cell receptor NKp30. Upon *MYC* inhibition in BL cell lines, NK cell-mediated degranulation via NKp30 was impaired [89].

In addition to its different effects on NK cells, *MYC* was found to impair innate immunity by its effects on tumor-associated macrophages (TAMs). In a mouse model of T-ALL overexpressing *Myc* and in DLBCL cell lines, *Myc* was associated with high CD47 expression on tumor cells, suppressing recruitment of macrophages and tumor phagocytosis [75,80]. Moreover, inactivation of *Myc* increased activation of innate immune cells and recruitment of macrophages to the tumor microenvironment [90].

### 3.3. Impact of MYC Overexpression on Apoptosis of Tumor Cells

Cytotoxic activity of T cells and NK cells is mainly established by induction of apoptosis (programmed cell death) in tumor cells [91]. During normal B cell development, MYC tightly regulates apoptosis of the developing B cells [92]. Elevated MYC levels can induce apoptosis, especially in cells that lack adequate levels of cytokines, growth factors or nutrients [93,94]. Overexpression of MYC induces apoptosis through activation of the ARF/MDM2/p53 pathway and altering the balance of pro- and anti-apoptotic BCL2 family members [95]. During oncogenic transformation pro-apoptotic functions of MYC are counterbalanced by the acquisition of cooperating gene alterations, such as *TP53* mutations and BCL2 overexpression [96,97]. Recently, it has been discovered that the *MYC*-related transcriptional repressor MNT reduces the level of pro-apoptotic BIM, thereby inhibiting pro-apoptotic functions of MYC [98]. Loss of BIM contributes to lymphomagenesis in a transgenic *Myc* mouse model [99]. Thus, during transformation the feedback loops involving proteins such as BLIMP1 and BCL6 that under normal circumstances would control MYC’s functions are abrogated [100]. The shift towards pro-survival signaling in MYC-transformed lymphoma cells can, therefore, contribute to immunochemotherapy resistance, including reduced sensitivity for T cell and NK cell-mediated apoptosis.

### 3.4. MYC-Dependent Regulation of Metabolism and Interplay with the Immune Microenvironment

The effects of *MYC* on metabolism regulation and its effect on tumor immune microenvironment regulation are summarized in Figure 2.

Increased expression of MYC promotes proliferation and growth of cells by inducing metabolic reprogramming and altering intermediary metabolism to match the enhanced demand for anabolic metabolites. Oncogenic levels of MYC promote high consumption of glucose as observed in most cancers, including BL [101,102,103], which correlates with the ability of MYC to regulate the expression of many genes involved in glycolysis, such as the glucose transporter *SLC2A1/GLUT1*, the glycolytic enzymes hexokinase 2 (*HK2*), triose-phosphate isomerase (*TPI*), enolase 1 (ENO1) and lactate dehydrogenase A (*LDHA*) [104,105,106]. MYC overexpression results in enhanced LDHA levels, which utilizes pyruvate as a substrate and converts it into lactate, thereby resulting in extracellular acidification [107,108]. Aerobic glycolysis is also driven by MYC-induced differential splicing and increased expression of the M2 isoform of pyruvate kinase (PKM2) [109].

The concept of “glutamine addiction”, a phenomenon also described for MYC overexpressing lymphomas [101,110], indicates the increased demand for glutamine as a source of carbon and nitrogen in cancer cells. By regulating expression of glutamine transporter genes *SCL1A5* (or ASCT2) and *SLC38A5* (or SNAT5), MYC facilitates transport of glutamine into the cell [111]. An important component of the glutaminolytic pathway is the enzyme glutaminase (*GLS*), which catalyzes the conversion of glutamine into glutamate. MYC upregulates the expression of *GLS* both directly and indirectly, through MYC-mediated repression of microRNA *(miR)-23* transcription, which in turn suppresses *GLS* translation [110,111]. To couple metabolic reprogramming with nucleotide synthesis, MYC regulates many genes involved in purine and pyrimidine synthesis [112,113,114].

Overall, the high energy demand of cancer cells together with a poor vascular flow and defective lymphatic drainage will result in a shortage of key metabolites and the abundance of waste products in the tumor microenvironment [115]. In the context of lymphomas with MYC overexpression, there is a tight competition for vital nutrients with the neighboring immune cells, and a negative impact of specific by-products generated by the MYC-expressing lymphoma cells. Lactate, the waste product of glycolysis and glutaminolysis, is considered an immunosuppressive metabolite that impairs immune surveillance of cytotoxic T cells and NK cells [116,117], recruits immunosuppressive regulatory T cells [118] and sustains the M2-like phenotype of TAMs [119,120]. Furthermore, glucose deprivation in the tumor microenvironment itself impairs T cell metabolic fitness by reducing T cell survival, IFN-γ production and upregulating PD-1 expression, leading to decreased anti-tumor immune responses [121]. Acidic conditions reduce the secretion of certain cytokines (IL-2, TNF-α, IFN-γ) by T cells, and upregulate CTLA-4 expression [122]. Other negative effects that have been reported in tumors as a consequence of competition with metabolites involve the differentiation of naïve T cells into immunosuppressive regulatory T cells (Tregs) and impaired NK cell function due to low glutamine levels [123,124,125].

## 4. Direct and Indirect MYC Modulating Drugs and Consequences for Immune Effector Cells

Despite wide efforts of studying *MYC* and its functions over the past decades, MYC has been considered an “undruggable” target for a long time. Moreover, narrow therapeutic windows and on- and off-target toxicities due to the complex interactome of MYC have been a major limitation for clinical testing of MYC inhibitors. Various attempts to target MYC in preclinical models and clinical studies are discussed in detail elsewhere [126,127,128,129]. Here, we will summarize some of the currently available drugs that directly and indirectly modulate MYC with specific emphasis on their possible impact on immune responses.

### 4.1. Bromodomain Inhibitors

Bromodomain inhibitors inhibit *MYC* transcription by competitively binding to bromodomain containing proteins (BRDT, BRD2, BRD3, BRD4), thereby preventing histone acetylation. JQ1 is a small molecule that competitively binds to BRD4 [130]. JQ1 is widely used to study the effects of MYC downregulation in lymphoid malignancies [75,76,80,131]. Bromodomain inhibitor analog, CPI-0610, has been well tolerated in patients with DLBCL in a phase I study [132]. However, due to its effects on BRD4 target genes, JQ1 inhibits MYC non-specifically [130]. For example, bromodomain inhibitors impair T cell survival, T cell activation and repress IFN-γ secretion [133,134,135]. On the other hand, T cells treated with JQ1 show enhanced anti-tumor immune responses in vivo [136]. The accompanying high amount of on- and off-target toxicities form a major limitation for clinical implementation of bromodomain inhibitors.

### 4.2. PI3K Inhibitors

Translation of *MYC* mRNA is partly regulated by mammalian target of rapamycin (mTOR), which is activated by AKT after its phosphorylation by phosphoinositide 3-kinase (PI3K) [137]. Kinases of the intracellular PI3K/AKT/mTOR signaling pathway regulate translation and posttranslational phosphorylation of MYC. The PI3K/AKT/mTOR pathway contributes to metabolic reprogramming at least partly via *MYC* [137]. Kinase inhibitors targeting the PI3K/AKT/mTOR pathway do not only serve as indirect MYC inhibitors, but also directly target cancer metabolism. Furthermore, the PI3K/AKT/mTOR signaling pathway is as an essential pathway in regulating cell metabolism, cell growth and cell differentiation, evenly well required in innate immune cells and effector and memory T cells [138,139]. Specifically, PI3K regulates Th1, Th2 and Th17 cell differentiation [140]. Furthermore, cytotoxic T cell trafficking and activation of cytotoxic T and NK cell effector functions are dependent on PI3K/AKT signaling [140,141], suggesting that these inhibitors may also play a role in modulating immune responses.

TGR-1202 selectively inhibits PI3Kδ in lymphoma, leading to decreased MYC protein levels and induced lymphoma cell death [142]. Idelalisib is the first FDA approved PI3K inhibitor and is now being tested in patients with DLBCL (NCT03576443). Fimepinostat (CUDC-907) is a combined HDAC and PI3K inhibitor that inhibits transcription and consequently downregulates MYC mRNA and protein levels [143]. Results of a phase I study show a tolerable safety profile with endurable anti-tumor responses in patients with MYC overexpressing DLBCL [144].

### 4.3. MYC/MAX Dimerization Inhibitors

Although inhibition of *MYC* at transcriptional and translational levels is effective, it is accompanied with multiple off-target effects and MYC protein could still retain a functional level that is regulated post-translationally by various microRNAs [145]. Therapeutic agents inhibiting MYC/MAX dimerization and drugs targeting MYC protein stability may represent more specific and effective therapies for MYC overexpressing lymphoid malignancies.

Preventing MYC/MAX dimerization can either be established by disruption of the MYC/MAX dimer or by inducing and stabilizing MAX/MAX homodimers. The earliest reported protein to prevent MYC/MAX dimerization is Omomyc, a MYC protein derivative from the bHLH-LZip domain with a four amino acid mutation that alters dimerization specificity of MYC [146]. The pharmacological and clinical application of Omomyc is currently being investigated [147,148]. Low molecular weight compound 10058-F4 is a common agent used to inhibit MYC/MAX heterodimer formation, inhibiting MYC’s biological and oncogenic functions [149,150]. Recently, a small molecule (KI-MS2-008) has been developed that stabilizes the MAX homodimer and prevents dimerization with MYC, resulting in reduced in vivo tumor volume [151]. Although MAX is currently the only known obligate dimerization partner of MYC and disrupting MYC/MAX dimerization significantly reduces MYC activity, MYC still retains some functional MAX-independent activity [152].

Recently developed small molecules MYCi 361 and MYCi 975 destabilize the MYC protein and consequently inhibit tumor growth [153]. A narrow therapeutic window and splenic and hepatic toxicities were reported in preclinical studies for MYCi 361, but MYCi 975 was better tolerable. Both MYC inhibitors displayed additional beneficial effects on enhanced immune cell infiltration in vivo [153]. Combined with anti-PD1 immunotherapy, these MYC inhibitors showed synergistic effects on tumor volume in an in vivo model of prostate cancer [153].

### 4.4. Immunomodulatory Drugs

Immunomodulatory drugs (IMiDs) feature, besides anti-proliferative effects, wide immunomodulatory properties, such as co-stimulation of T cells, enhancement of NK cell activity and increased IL-2 and IFN-γ secretion [154,155]. Lenalidomide, a second generation orally available IMiD, downregulates MYC via interferon regulatory factor 4 (IRF-4). IRF-4 is downregulated after binding of IMiDs to Cereblon and consequent ubiquination and degradation of its substrate proteins [156,157]. Addition of lenalidomide to standard chemotherapy in patients with MYC overexpressing DLBCL is safe and results in comparable responses to outcomes of intensified chemotherapy regimens [158]. Similar studies investigating lenalidomide in combination with standard chemotherapy in patients with DLBCL are ongoing (DA-EPOCH; NCT02213913, or R-CHOP; NCT04164368).

## 5. Immunotherapies in DLBCL, with Special Emphasis on Patients with MYC Overexpression

In this section, we review current clinical outcomes of DLBCL patients treated with various immunotherapeutic strategies. Only a limited number of studies were specifically designed for lymphomas with MYC overexpression. Most studies reported outcome of patients with MYC overexpression and MYC normal expression as subgroups. Figure 3 summarizes current immunotherapeutic treatment strategies and their possible mechanism of action in MYC overexpressing DLBCL.

### 5.1. T Cell Engaging and Modulating Therapies

Bispecific T cell engager antibodies targeting T cells to a tumor antigen (CD3xCD19 or CD3xCD20) [159], are currently a promising treatment strategy in lymphoid malignancies [160,161,162]. Bispecific antibodies targeting other antigens (anti-CD47/CD19 TG-1801 NCT03804996, PD-L1/CD137 (4-1BB) NCT03922204) are also currently being investigated in various groups of B-NHL in ongoing clinical trials. In a phase II study of blinatumomab (CD3xCD19), one patient with a TH HGBL was included and achieved a complete response [161].

Immune checkpoint inhibitors block the interaction of immune checkpoint receptors, such as PD-1 or CTLA-4, with their ligands, thereby enhancing the amplitude of T cell activation [163]. Although effective in patients with classical Hodgkin lymphoma [164,165], patients with DLBCL show inferior outcomes towards nivolumab monotherapy as compared with standard treatment. The numbers of patients included are too small to separately evaluate the response of MYC overexpressing patients [166]. Currently, phase II clinical studies are ongoing to investigate the effect of additional PD-1 checkpoint inhibitor nivolumab to standard chemotherapy in MYC overexpressing HGBL (NCT03620578 specifically designed for previously untreated HGBL, NCT03038672 and NCT03749018 including HGBL). First-line administration of pembrolizumab, another PD-1 immune checkpoint inhibitor, showed promising results in combination with R-CHOP in untreated DLBCL patients, irrespective of MYC protein expression [167]. Anti-PD-L1 durvalumab is currently being investigated in combination with R-CHOP with or without lenalidomide in previously untreated DLBCL (NCT03003520).

CD47 blocking immune checkpoint inhibitor Hu5F9-G4 is well tolerated and shows potential effective results in patients with relapsed or refractory DLBCL [168]. Although CD47 is pre-clinically correlated with MYC expression [75], consequences of MYC status on clinical outcome with anti-CD47 antibodies are not yet known [168].

### 5.2. Cell-Based Immunotherapies

In allogeneic stem cell transplantation donor T cells recognize and eradicate lymphoma cells, achieving the immunologic “graft versus tumor” effect. Prospective studies evaluating the role of allogeneic stem cell transplantation for patients with MYC overexpression are lacking. The two largest retrospective studies for relapsed DLBCL patients show different results; in one study, long term survival was lower in MYC overexpressing patients [169]. Another study did not observe any differences in 4-year survival between DLBCL patients with or without MYC overexpression [170].

Over the past decades, the field of cell-based immunotherapy has evolved rapidly with the introduction of T cells engineered to express a chimeric antigen receptor (CAR) towards a specific target [171]. Recently, CAR T cells targeting CD19 on B cells have been developed [171]. One preclinical study showed a positive correlation between MYC and CD19 expression. [172], suggesting beneficial effects of CD19 CAR T cell therapy in HGBL patients.

Three independent clinical trials investigated the effect of different CD19 targeting CAR T products in DLBCL and reported outcomes of patients with MYC overexpressing lymphomas as subgroups [173,174,175]. In summary, no differences in MYC overexpressing patients compared with patients with normal MYC expression were observed regarding response and survival with a median follow up of 28, 16 and 18 months, respectively [173,174,175]. These results suggest that CAR T cell therapies might overcome the negative effects of MYC overexpression. Efficacy of novel CAR T cells targeting other antigens, such as CD22, CD30 and CD79b [176,177,178,179,180,181] and innate immunity-mediated CAR NK cells [182] on MYC overexpressing lymphomas are under investigation.

## 6. Conclusions and Perspectives

MYC regulates several aspects of normal immune cell development and can drive lymphomagenesis upon oncogenic transformation. In this review, we described the complex role of MYC overexpression on malignant B cells in the regulation of anti-tumor immune responses and lymphoma immune evasion. In lymphoid malignancies, most studies demonstrate that MYC overexpression negatively affects antigen presentation, T cell recognition and T cell and NK cell-mediated cytotoxicity, resulting in lymphoma immune evasion. These findings are largely comparable with the effects of MYC on anti-tumor immune responses across different types of cancer [183]. Therefore, MYC modulation has the potential to enhance anti-tumor immune responses. However, we also discussed the consequences of *MYC* modulation for immune effector cells. *MYC* is important for differentiation and proliferation of T cells [28,184,185] and for glycolysis, metabolic reprogramming and cell cycle re-entry of activated T cells [26,186,187]. In NK cells, MYC activates receptors on NK cells that regulate NK cell activation and tolerance [188] and MYC drives NK cell expansion during antitumor immunity [189]. As a consequence, beneficial effects of MYC downregulation in malignant target cells may be overshadowed by the effects of MYC downregulation in immune effector cells, reducing their cytotoxic anti-tumor immune responses. Due to the complex effects of MYC deregulation on immune responses, important unanswered questions, especially regarding the effects of MYC on PD-L1, NK cells and the tumor microenvironment remain.

Of all immunotherapeutic strategies, CAR T cell therapy seems to circumvent the negative effects of MYC on immune effector cells. Indeed, three studies demonstrated no inferior outcomes for MYC overexpressing DLBCL patients that received CD19 CAR T cell therapy [173,174,190].

To guide future treatment approaches, further research is needed to elucidate the role of *MYC* on immune responses in MYC overexpressing lymphomas. So far, in most clinical studies reviewed here, the outcome of patients with and without MYC overexpression was not reported separately, which we would highly recommend doing in future studies. At present, for patients with MYC overexpression, cellular immunotherapies seem most promising. Ideally, these therapies are combined with MYC modulators that improve immunogenicity and facilitate apoptosis. However, current available MYC modulators compromise effector cell function. We expect that more insight into the downstream MYC signaling events playing key roles in immune escape, may facilitate the development of successful cellular immunotherapy approaches that could more specifically exploit the MYC-induced immune escape.

## Figures and Tables

**Figure 1 cancers-12-03052-f001:**
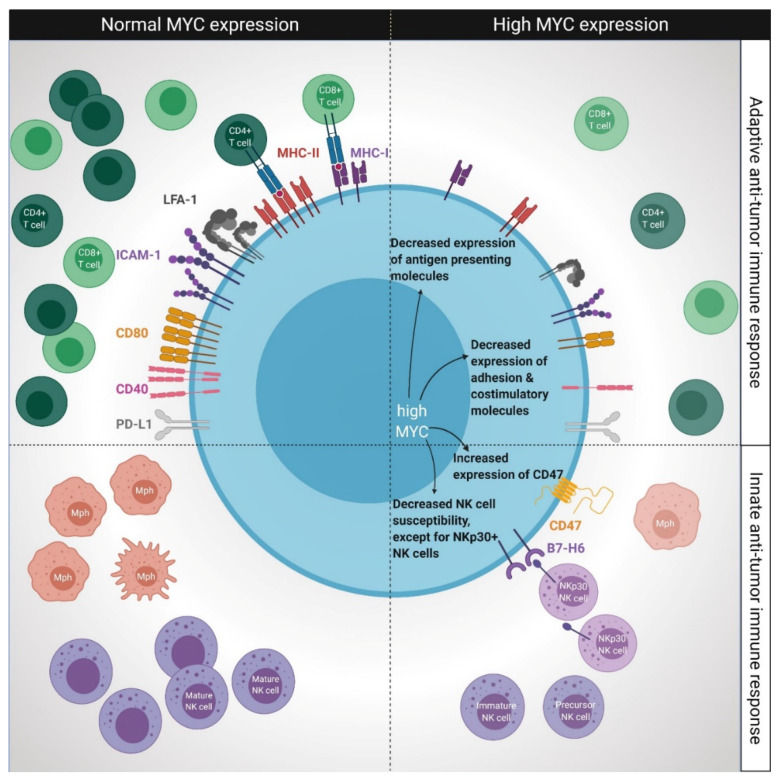
MYC overexpression is associated with reduced T cell-mediated anti-tumor immune responses and impaired innate anti-tumor immune responses by macrophages and NK cells, but not NK cells with the NKp30 receptor. Top left: simplified summary of adaptive immune responses towards malignant cells with normal MYC expression. Top right: MYC overexpression decreases expression of MHC class I and class II molecules, adhesion and costimulatory molecules. MYC overexpression is associated with reduced cytotoxic T cell responses. Different outcomes on the correlation between MYC overexpression and expression of immune checkpoint programmed death-ligand 1 (PD-L1) have been described. Bottom left: simplified summary of innate immune responses towards malignant cells with normal MYC expression. Bottom right: MYC overexpression reduces NK cell amounts, but not cytotoxicity induced via NKp30 NK cells. MYC overexpression induces expression of immune checkpoint CD47, preventing macrophage (Mph)-induced phagocytosis.

**Figure 2 cancers-12-03052-f002:**
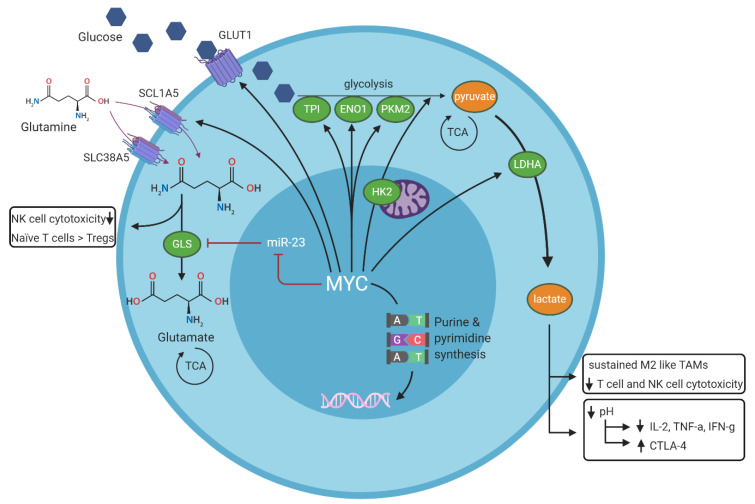
Summary of MYC-dependent regulation of cell metabolism and its effects on the tumor immune environment. MYC stimulates aerobic glycolysis (the conversion of glucose into pyruvate) via glycolytic enzymes TPI, ENO1, PKM2 and HK, and MYC stimulates anaerobic glycolysis (the conversion of glucose into lactate) via LDHA. MYC stimulates glutaminolysis (the conversion of glutamine into glutamate) by inhibiting microRNA-23. Waste products of all three pathways contribute to decreased effector cell cytotoxicity in the tumor immune environment. Green ovals represent enzymes, dark red lines represent inhibitory signals and black arrows represent stimulatory signals. TPI = triose-phosphate isomerase, ENO1 = enolase 1, PKM2 = M2 isoform of pyruvate kinase, TCA = tricarboxylic acid (TCA) cycle, LDHA = lactate dehydrogenase A, TAMs = tumor associated macrophages, GLS = glutaminase, HK2 = hexokinase 2. GLUT1 = glycose transporter 1, SCL1A5 = solute carrier family 1 member 5 (or alanine serine cysteine transporter 2, ASCT2), SLC38A5 = solute carrier family 38 member 5 (or sodium-coupled neutral amino acid transporter 5, SNAT5).

**Figure 3 cancers-12-03052-f003:**
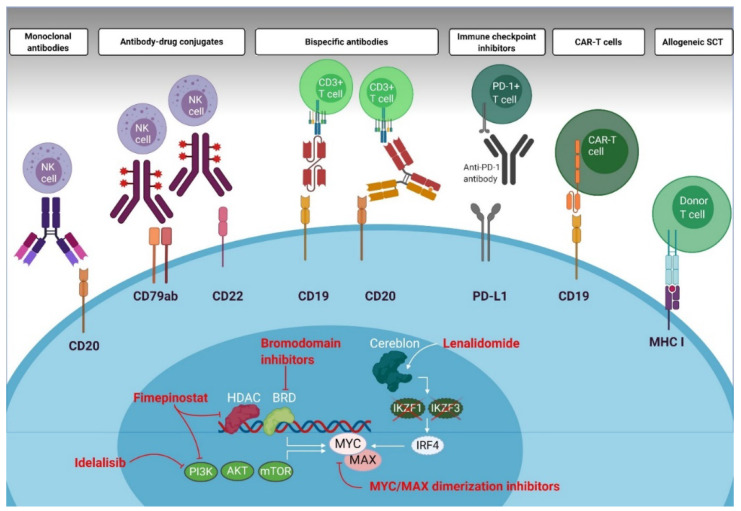
Immunotherapeutic treatment strategies and clinically available MYC modulators in lymphoid malignancies. Antibody-mediated therapies include monoclonal antibodies, antibody-drug conjugates, bispecific antibodies and immune checkpoint inhibitors. Cellular therapies include CAR-T cell therapy and allogeneic stem cell transplantation. Clinically available MYC modulators are lenalidomide (which reduces MYC protein by stimulating Cereblon, subsequently degrading IKZF1 and IKZF3, followed by downregulation of IRF4), idelalisib (PI3K inhibitor) and fimepinostat (combined PI3K and HDAC inhibitor). Preclinical MYC modulators are bromodomain (BRD) inhibitors (e.g., JQ1) and MYC/MAX dimerization inhibitors.

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
