# Peer review of "Impact of MYC on Anti-Tumor Immune Responses in Aggressive B Cell Non-Hodgkin Lymphomas: Consequences for Cancer Immunotherapy"

_cancers, 2020, doi:10.3390/cancers12103052_

Round 1

Reviewer 1 Report

The review is well written. The  unanswered questions are not sufficiently emphasized, maybe they can be summarized at the end at the end. The two figures are informative and well elaborated The roles of MYC are multiple and the effects of MYC deregulation are complex, particularly those concerning the immune response. So a review on this topic is welcome.. I have a few comments

1.- The title refers to non-Hodgkin lymphomas (NHL) , but most of the  Introduction deals with DLBC, which is most prevalent NHL,  without mentioning other types except Burkitt lymphoma at the end of the section. I would recommend initiating this part with a brief (a couple of lines) description of the different types of NHL and their overall incidence

2.- In line 36: “About 5-15% of all DLBCLs harbor translocations that affect the MYC” a seems contradictory with “In about 30% of the DLBCL patients, MYC is the only translocation (single hit (SH) DLBCL),”. This latter seems to refer to 30% among those with MYV translocations but it may be written clearer

3.- Lines 123-5: “where switching MYC off resulted  in CD40 downregulation”: mention what CD40 is (a TNF receptor) so the reader can tell the importance of this MYC-mediated downregulation. Same with CD80

4.- line 209: “…its effect on tumor immune microenvironment regulation are summarized in Figure 24. Materials and Methods”. They mean Figure 2 and there are no Materials and Methods here

5.- lines 224 and 230: They should mention other genes encoding enzymes of the glycolylitic pathway that are also MYC target genes: Enolase, TPI, PK. Importantly, the glutamine transporter  SCL1A5 is also MYC target gene. This should also be drawn in Figiure  2.

6.- Section 4.3. MYC/MAX dimerization inhibitors. The OmoMyc should be included in this section. This peptide is so far the best-positioned candidate to be used in clinics as anti-Myc drug

7.-There is a published review on similar topic: The MYC oncogene is a global regulator of the immune response (Casey et al,. Blood 2018), that is not cited in this manuscript.

Reviewer 2 Report

The manuscript by de Jonge et al. is an exceptionally insightfully and thorough review of the role of MYC in diffuse large B cell lymphoma. The paper is very well written and presents balanced and highly useful information of the field. Literature citations are carefully chosen and provide solid documentation. Illustrations are concise and graphically pleasing. This review is equally valuable for consultation, guidance and reference. That said, we have to admit that we do not understand the workings of MYC in cancer. Most probably, this will involve multiple pathways. We have pieces of a mosaic but no way to construct a unified hypothesis.

Reviewer 3 Report

Vera de Jonge and collaborators have reviewed the impact of MYC on anti-tumor immune responses in B-cell non-Hodgkin lymphomas. They present a very comprehensive and updated review. They include a brief introduction, a main part addressing the role of MYC overexpression on the immune system in lymphoid malignancies and a final part discussing direct and indirect MYC modulating drugs.

Overall, I understand that the paper summarizes pretty well the current knowledge in the field. Many previous reviews are focused in the proliferating effects of MYC but the modulation of immune system by MYC is definitively a less well know role. I believe that this review will be of help for investigators and physicians who are working in B-cell NHL.

The only minor point is regarding the section of modulating drugs in which there are other clinical studies evaluating the role of MYC interfering agents in DLBCL but there are other reviews addressing this (for instance: https://doi.org/10.1002/ajh.25460).

I have seen only a minor typing mistake: line 209: “Figure 24.Material and Methods”??? I understand this is Figure 2.

In Figure 3, I will change “MYC inhibitors” by “MYC modulating agents or drugs”, at least in line 373 and line 376.
